# Bio-Pulsed Stimulation Effectively Improves the Production of Avian Mesenchymal Stem Cell-Derived Extracellular Vesicles That Enhance the Bioactivity of Skin Fibroblasts and Hair Follicle Cells

**DOI:** 10.3390/ijms232315010

**Published:** 2022-11-30

**Authors:** Ju-Sheng Shieh, Yu-Tang Chin, Hsien-Chung Chiu, Ya-Yu Hsieh, Hui-Rong Cheng, Hai Gu, Fung-Wei Chang

**Affiliations:** 1Center for Health Policy and Management Studies, Nanjing University, 22 Hankou Road, Nanjing 210093, China; 2Department of Periodontology, School of Dentistry, Tri-Service General Hospital, National Defense Medical Center, Taipei City 11490, Taiwan; 3School of Nutrition and Health Science, Taipei Medical University, Taipei City 11031, Taiwan; 4Department of Obstetrics and Gynecology, Tri-Service General Hospital, National Defense Medical Center, No.325, Sec.2, Chenggong Rd., Neihu District, Taipei City 11490, Taiwan

**Keywords:** mesenchymal stem cell, small extracellular vesicle, exosome, bio-pulse, avian, human skin fibroblast, human follicle dermal papilla cell, aesthetics

## Abstract

Mesenchymal stem cell (MSC)-derived extracellular vesicles (exosomes) possess regeneration, cell proliferation, wound healing, and anti-senescence capabilities. The functions of exosomes can be modified by preconditioning MSCs through treatment with bio-pulsed reagents (*Polygonum multiflorum* Thunb extract). However, the beneficial effects of bio-pulsed small extracellular vesicles (sEVs) on the skin or hair remain unknown. This study investigated the in vitro mechanistic basis through which bio-pulsed sEVs enhance the bioactivity of the skin fibroblasts and hair follicle cells. Avian-derived MSCs (AMSCs) were isolated, characterized, and bio-pulsed to produce AMSC-sEVs, which were isolated, lyophilized, characterized, and analyzed. The effects of bio-pulsed AMSC-sEVs on cell proliferation, wound healing, and gene expression associated with skin and hair bioactivity were examined using human skin fibroblasts (HSFs) and follicle dermal papilla cells (HFDPCs). Bio-pulsed treatment significantly enhanced sEVs production by possibly upregulating *RAB27A* expression in AMSCs. Bio-pulsed AMSC-sEVs contained more exosomal proteins and RNAs than the control. Bio-pulsed AMSC-sEVs significantly augmented cell proliferation, wound healing, and gene expression in HSFs and HFDPCs. The present study investigated the role of bio-pulsed AMSC-sEVs in the bioactivity of the skin fibroblasts and hair follicle cells as mediators to offer potential health benefits for skin and hair.

## 1. Introduction

Mesenchymal stem cells (MSCs) have considerable potential for application in tissue engineering and regenerative medicine because of their ability to regulate inflammation, inhibit apoptosis, promote angiogenesis, and reinforce the growth and differentiation of local stem and progenitor cells [1]. MSCs were first discovered in the bone marrow, and bone marrow MSCs (BMSCs) have a high proliferative ability and can differentiate into adipocytes, osteoblasts, myoblasts, and neurons [2]. However, the number of BMSCs and their proliferation and differentiation abilities substantially decrease with aging. In recent years, researchers have attempted to search for MSCs in other tissues, such as the muscles, amniotic fluid, umbilical cord blood, and adipose tissue [3].

The dermis mainly contains differentiated cells, including fibroblasts, which typically participate in scar tissue formation during skin repair [4]. Thus, the dermis is often used as a negative control in stem cell studies [5]. In the past two decades, considerable progress has been made in research on dermis-derived MSCs, with studies focusing on their separation, culture, and induced-differentiation into osteoblasts, adipocytes, and ectodermal cell types [2]. Because of the easy accessibility of dermis-derived MSCs, they have become an ideal cellular source for tissue engineering [6,7]. Most studies have focused on stem cells derived from humans, mice, rabbits, and other mammals, whereas few studies have examined stem cells from poultry. As a model animal, the chicken possesses abundant dermal tissues. Furthermore, chicken is an avian species that is crucial in the global economy [2]. In this study, we applied chicken dermis-derived MSCs not only to obtain a stable source of cells but also to avoid embryo retrieval surgery from mothers (mammals). Furthermore, according to the Cosmetic Products Regulations in Europe, China, and Taiwan, the use of cells, tissues, or products of human origin in cosmetic products is prohibited. AMSCs may be an ideal cell source in tissue engineering and industrial applications, including the production of extracellular vesicles or exosomes for aesthetic applications or use in cosmetic products in the future.

Intercellular communication, a highly conserved cell process, occurs through either direct cell-to-cell contact or paracrine secretion [8]. MSCs exhibit their regeneration potential in a paracrine manner through their secretome [9]. The secretome of MSCs contains growth factors, cytokines, and extracellular vesicles (EVs), including exosomes. These EVs contain various macromolecules that alter the destiny of recipient cells through paracrine signaling. Although EVs are secreted by almost all cell types, MSCs produce the highest amount of EVs [10]. Exosomes are nanosized vesicles that are released from various cells, including fibroblasts, macrophages, tumor cells, and MSCs, and they can be found in the amniotic fluid, milk, urine, ascites fluid, blood, cerebrospinal fluid, and saliva [11]. Exosomes are the main subtype of EVs, with a diameter ranging from 30 to 150 nm. Exosomes include various biological components, including miRNAs, proteins, lipids, and mRNAs, as cargo [12]. The production and secretion processes of exosomes involve three main steps: (1) creation of endocytic vesicles through the invagination of the plasma membrane, (2) production of multivesicular bodies (MVBs) upon the inward budding of endosomal membranes, and (3) incorporation of the MVBs with the plasma membrane and the release of vesicular contents termed as exosomes [13]. Therefore, the transfer of the contents of exosomes to recipient cells leads to the modification of physiological cells [14]. Exosomes play a crucial role in intercellular communication, and specific cell-derived exosomes trigger the directed differentiation of stem cells [15]. The composition of exosomes can be modified by preconditioning MSCs using various factors, including hypoxia, cytokines, growth factors, and pharmacological agents, or by growing them under three-dimensional culture conditions [16,17]. Studies also reported that MSC-derived exosomes enhance the biological properties of fibroblasts by encapsulated miRNAs, including antisenescence, anti-oxidative stress, and wound healing [18,19,20,21,22].

Cell priming (also referred to as licensing or preconditioning) with proinflammatory mediators is among the first reported approaches [23,24]. Cell priming (here called biochemical pulse pattern, abbreviated as bio-pulsed) involves preparing cells for lineage-specific differentiation or conferring the cells with specific functions through cell activation, molecular signaling activation, genetic or epigenetic modifications, and morphology/phenotype alterations. Recently, studies have reported that the composition of the MSCs’ extracellular vesicles or exosomes is dependent upon the microenvironment in which they thrive, and hence it can be altered by pre-conditioning or priming the MSCs during in vitro culture [25]. MSCs were primed with pro-inflammatory cytokines or hypoxia; primed-MSCs derived extracellular vesicles or exosomes that were isolated and reported with special functions, such as anti-inflammation or better therapeutic efficiency [26,27].

*Polygonum multiflorum* (PM) Thunb is a traditional Chinese medicinal herb that has been widely used for thousands of years, particularly for treating age-associated diseases [28]. Tetrahydroxystilbene glucoside (TSG) is a major bioactive compound in PM [29]. Various studies have focused on the numerous pharmacologic activities of TSG, including anti-inflammatory [30], anti-cancer [31], neuroprotective [32], and cardioprotective [33] activities. Moreover, studies have investigated the anti-thinning effect of derma on the skin [34] and hair growth [35]. In our previous study, we found that PM extract stimulated dental pulp stem cell-derived conditioned medium enhanced the activity and anti-inflammation properties of cells [36].

We hypothesized that bio-pulsed AMSC-derived sEVs may have better beneficial efficacy and potential benefits for skin fibroblasts and hair follicle cells. The aim of this study is to investigate the effects of bio-pulsed treatment using the PM extract as the bio-pulsed reagent on chicken embryonic MSCs (avian-derived MSCs (AMSCs)). In addition, we established procedures for the isolation, purification, extraction, and lyophilization of bio-pulsed AMSC-derived small extracellular vesicles (sEVs). Furthermore, we examined the characteristics of bio-pulsed AMSC-derived sEVs and the better beneficial effects of their cell activity on skin fibroblasts and hair follicle cells.

## 2. Results

### 2.1. Characterization of AMSCs and Bioactivity of AMSCs Treated with Bio-Pulsed Reagent

The primary AMSCs isolated from embryonic tissues adhered to culture flasks and exhibited fibroblast-like morphology (Figure 1A). We examined the MSC markers of AMSCs through qRT-PCR. The results of qRT-PCR revealed that passage 1 and passage 7 of AMSCs expressed CD44, CD71, and CD73, but not CD34 (a hematopoietic cell marker). In addition, the expression pattern of MSC markers was similar (Figure 1A). The bio-pulsed reagent was the PM extract, and we investigated its dose effect on the proliferation of the AMSCs and the gene expression of *MKI67*, *PCNA*, and *RAB27A*. Treatment of the AMSCs with the PM extract for 4 days significantly enhanced cell proliferation (Figure 1B). RAB27A, a Rab GTPase, regulates the release of exosomes [37,38,39]. After the treatment of the AMSCs with the PM extract for 3 days, not only the expression of *MKI67* and *PCNA* (the cell proliferative genes) but also the expression of *RAB27A* were significantly upregulated, especially after treatment with 10 μg/mL PM extract (Figure 1C). Thus, we used 10 μg/mL of the PM extract for the subsequent production of bio-pulsed AMSC sEVs.

### 2.2. Quantity and Quality of AMSC-sEVs Were Enhanced by Bio-Pulsed Treatment

To examine the morphology, quantity, and quality of the sEVs produced after bio-pulsed treatment and lyophilization, we performed TEM examination and TRPS measurements. The morphology of the sEVs did not change after lyophilization (Figure 2A). The particle size distribution and number of the sEVs differed after bio-pulsed treatment; however, only the number of sEVs after the bio-pulsed treatment was significantly different to untreated AMSC-sEVs. The particle size distribution and number of bio-pulsed AMSC-sEVs were smaller and lower than those of the untreated AMSC-sEVs (Figure 2B). Furthermore, the findings of Western blotting revealed the expression of the general surface markers CD9, CD63, CD81, and TSG101 on the bio-pulsed AMSC-sEVs, but the contamination marker, calnexin, was not expressed (Figure 2C).

In order to examine the residual protein contained in the lyophilized sEVs, we determined the presence of protein contamination in the lyophilized sEVs powder. Both the control AMSC CM and bio-pulsed AMSC CM contained a high protein concentration (Figure 3A). However, both the lyophilized control and bio-pulsed AMSC-sEVs powders did not contained residual proteins (Figure 3A). To investigate whether the miRNAs contained in bio-pulsed AMSC-sEVs are degraded or were resistant to the presence of RNase, lyophilized bio-pulsed AMSC-sEVs were reconstituted and treated with RNaseA (100 ng/mL) before RNA extraction. The small RNA profiles of the bio-pulsed AMSC-sEVs with or without RNaseA treatment did not exhibit the degradation of RNA, showing that exosomal small RNAs were protected by sEVs (Figure 3B). Further, exosomal proteins and RNAs from both the lyophilized control and bio-pulsed AMSC-sEVs were examined through lysis and extraction to validate their quality. The results revealed that the bio-pulsed AMSC-sEVs contained significantly more exosomal proteins and RNAs than the control AMSC-sEVs (Figure 3C,D).

### 2.3. Bio-Pulsed AMSCs-sEVs Enhanced Cell Proliferation and Wound Healing

To investigate the effect of the bio-pulsed AMSC-sEVs on cell proliferation, the HSFs and HFDPCs were treated with the AMSC-sEVs and various concentrations of the bio-pulsed AMSC-sEVs for 96 h. The results demonstrated that the AMSC-sEVs enhanced the proliferation of the HFDPCs, but not the HSFs (Figure 4A). However, the bio-pulsed AMSC-sEVs augmented the proliferation of both the HSFs and HFDPCs in a dose-dependent manner (Figure 4A). Furthermore, the bio-pulsed AMSC-sEVs, especially at the concentration of 70 μg/mL, enhanced wound healing in the HSFs within 24 h (Figure 4B).

### 2.4. Bio-Pulsed AMSCs-sEVs Enhanced Collagen Expression and Anti-Inflammation

Studies have reported that EVs have helpful effects on skin, including anti-inflammation, regeneration, and ECM production [40,41,42]. To investigate the effect of the bio-pulsed AMSC-sEVs on collagen and elastin expression, the HSFs were treated with the AMSC-sEVs and various concentrations of the bio-pulsed AMSC-sEVs for 48 h. The expression of *COL1A1*, *COL3A1*, and *ELN* was significantly enhanced in the HSFs after treatment with the AMSC-sEVs. However, treatment with the bio-pulsed AMSC-sEVs more significantly upregulated the expression of the aforementioned genes, especially at the dose of 70 μg/mL (Figure 5A). We examined the anti-inflammatory effect of the bio-pulsed AMSC-sEVs on HSFs. The expression of *IL-1β*, *TNF-α*, and *IL-6* was enhanced in the HSFs after treatment with 1 μg/mL of *E. coli* LPS; this effect was significantly ameliorated by the AMSC-sEVs and bio-pulsed AMSC-sEVs (Figure 5B).

### 2.5. Bio-Pulsed AMSCs-sEVs Enhanced the Activity of Hair Follicle Cells

We investigated the effect of the bio-pulsed AMSC-sEVs on hair follicle activity-associated genes, proliferative genes, and apoptosis genes. The HFDPCs were treated with the AMSC-sEVs and various concentrations of the bio-pulsed AMSC-sEVs for 48 h. The expression of the follicle activity-associated genes *RAC1* and *IGF-1* were significantly enhanced in the HFDPCs after treatment with the AMSC-sEVs; however, the expression of these genes was more significantly upregulated after treatment with the bio-pulsed AMSC-sEVs, especially at the dose of 350 μg/mL (Figure 6A). The expression of the cell proliferative genes *CCND1* and *PCNA* was significantly increased in the HFDPCs after treatment with the AMSC-sEVs; however, the expression of these genes was more significantly upregulated after treatment with the bio-pulsed AMSC-sEVs, especially at the dose of 350 μg/mL (Figure 6B). The expression of the apoptosis genes *CASP2* and *TP53* (p53) decreased after treatment with the AMSC-sEVs and diminished following treatment with the bio-pulsed AMSC-sEVs, especially at the dose of 350 μg/mL (Figure 6B).

## 3. Discussion

In this study, we successfully isolated the embryonic tissues of 10-day-old chick embryos, which are younger and more suitable for isolating embryonic MSCs (Figure 1A). Multilineage differentiation of stem cells is the most remarkable characteristic for homotransplantation. Because of easy accessibility, AMSCs have become an ideal cell source in tissue engineering and industrial applications, including the production of extracellular vesicles or exosomes for aesthetic applications or use in cosmetic products. According to the Cosmetic Products Regulations in Europe, China, or Taiwan (e.g., Regulation (EC) No 1223/2009 of the European Parliament and European Council), the use of cells, tissues, or products of human origin in cosmetic products is prohibited. This is the reason why we selected AMSCs to study. MSCs isolated from different tissue sources exhibit varying cellular composition, lineage-specific differentiation potential, and self-renewal capabilities [43]. MSCs have been widely explored in cell-based therapies because of their excellent anti-inflammatory, immunomodulatory, and regenerative properties [44], which involve both paracrine and cell-to-cell contact mechanisms. The paracrine effect depends on the MSC secretome, which involves numerous bioactive molecules, such as growth factors, cytokines, chemokines, and microvesicles/exosomes carrying proteins and miRNAs to target cells [44,45].

MSC activities and survival are considerably affected by in vivo and in vitro biological, biochemical, and biophysical factors [24] through reciprocal interactions between cells, the extracellular matrix (ECM), soluble bioactive factors, and EVs. The modulation of biological, biochemical, and biophysical factors can affect the fate, lineage-specific differentiation, activities, and functions of MSCs and enhance their therapeutic potential [46,47,48]. Cell priming with pro-inflammatory mediators is among the first reported approaches [23,24]. Cell priming involves preparing cells for lineage-specific differentiation or conferring the cells with specific functions through cell activation, molecular signaling activation, genetic or epigenetic modifications, and morphology/phenotype alterations. This concept is commonly used in the field of immunology and has been adapted for stem cells [49]. In this study, we treated AMSCs with the PM extract, a bio-pulsed reagent. Treatment with the PM extract enhanced the proliferation of the AMSCs, especially at the dose of 10 μg/mL (Figure 1B). Moreover, the PM extract upregulated the expression of *MKI67*, *PCNA* (proliferative markers), and *RAB27A*, particularly at the dose of 10 μg/mL (Figure 1C). Because RAB27A regulates the release of exosomes [37,38,39], the particle number of the bio-pulsed AMSC-sEVs was considerably more than that of the control AMSC-sEVs (Figure 2B). On the basis of our experimental findings, we used a PM concentration of 10 μg/mL in the subsequent experiments. In addition, we confirmed that exosomal small RNAs in bio-pulsed-sEVs can be protected after RNase treatment (Figure 3B). The quantity of exosomal proteins and RNAs was higher in the bio-pulsed AMSC-sEVs (Figure 3C,D). Researchers have reported that lyophilization is promising for the storage of EV products [50,51]. They concluded that freeze-dried EVs/exosomes could be stored at −20 °C and were ready-to-use. In general, the lyophilization technique has been considered as a cost-saving strategy for EV preservation and may be used to extend the shelf life of EVs without affecting their particle morphology and contents when stored at RT [50]. However, the detailed mechanism underlying this finding warrants further investigation.

The bio-pulsed AMSC-sEVs exhibited prominent bioactivity for the skin and hair. Although skin and hair consist of more than HSFs and HFDPCs, many studies have applied HSFs and HFDPCs to examine the functions of exosomes on skin and hair [20,22,52]. In this study, we selected HSFs and HFDPCs as model cells to examine the potentially beneficial efficacy of bio-pulsed AMSC-sEVs on skin and hair. Compared with the control AMSC-sEVs, the bio-pulsed AMSC-sEVs not only enhanced the proliferation of the HSFs and HFDPCs (Figure 4A) but also augmented wound healing in the HSFs (Figure 4B). Furthermore, the bio-pulsed AMSC-sEVs upregulated the expression of *COL1A1*, *COL3A1*, and *ELN*, which are crucial in skin remodeling, reconstruction, flexibility, and elasticity (Figure 5A). Skin aging is characterized by the fragmentation of collagen fibrils and a reduction in collagen type I and III synthesis [53]. Thus, collagen is among the most vital proteins of the ECM that imparts a fuller and younger appearance to tissues, and elastin is essential for skin elasticity [54]. Our findings revealed that the effects of bio-pulsed AMSC-sEVs on skin fibroblasts are consistent with previous results that EVs secreted by TGFβ-stimulated umbilical cord mesenchymal stem cells on skin fibroblasts promoted fibroblast migration and ECM protein production [55].

Moreover, in this study, the bio-pulsed AMSC-sEVs exhibited a remarkable anti-inflammatory effect on HSFs, significantly ameliorating the *E. coli* LPS-induced expression of the proinflammatory genes *IL-1β*, *TNF-α*, and *IL-6* (Figure 5B). Our bio-pulsed AMSC-sEVs by PM extract showed similar anti-inflammatory effects with curcumin reinforced-MSC-derived exosomes [56]. *RAC1* and *IGF-1* are crucial in hair follicle activity and hair regeneration [57,58,59,60]. The AMSC-sEVs significantly upregulated the expression of *RAC1* and *IGF-1* in the HFDPCs; however, the bio-pulsed AMSC-sEVs more significantly enhanced their expression (Figure 6A). Furthermore, the bio-pulsed AMSC-sEVs enhanced cell proliferation by upregulating the expression of the proliferation-associated genes *CCND1* and *PCNA* and inhibiting the expression of the apoptosis-associated genes *CASP2* and *TP53* (Figure 6B). In this study, we applied chicken dermis-derived MSCs to produce non-human origin sEVs and bio-pulsed stimulation to obtain unusual sEVs. We tried to prove that bio-pulsed AMSC-sEVs can be the potential mediators of health benefits in aesthetic medicine, and they may be commoditized for cosmetics.

In conclusion, the present study highlights the role of bio-pulsed AMSC-sEVs in the bioactivity of skin fibroblasts and hair follicle cells. The PM extract as the bio-pulsed reagent stimulates AMSCs to produce various sEVs, which exhibit remarkable and beneficial effects on skin fibroblasts and hair follicle dermal papilla cells. Thus, the bio-pulsed AMSC-sEVs exhibited bioactivity with potential benefits for skin fibroblasts and hair follicle cells.

## 4. Materials and Methods

### 4.1. Isolation and Culture of AMSCs

The experimental use of fertile chick eggs was approved by the Animal Health Research Institute, Council of Agriculture (New Taipei City, Taiwan). These eggs were obtained from the same institute. We used the isolation and culture procedures described in a previous study [2], with minor modifications. We removed the head, wings, feet, and body cavity content of 10-day-old chick embryos and then isolated their embryonic tissues. The isolated embryonic tissues were cut into approximately 0.5-cm^2^ pieces and then digested with 0.25% trypsin (Thermo Fisher Scientific Inc., Waltham, MA, USA) for approximately 20 min. Subsequently, enzymatic activity was neutralized using fetal bovine serum (FBS; Thermo Fisher Scientific Inc.). The digested tissues were passed through a 70-μm mesh filter and then centrifuged at 500× *g* for 10 min at room temperature. The supernatant was discarded, and the pellet was resuspended in Dulbecco’s modified Eagle’s medium (DMEM) supplemented with a high glucose concentration (4.5 g/L) (Thermo Fisher Scientific Inc.) and 5% FBS. The viability of AMSCs was determined using the trypan blue exclusion method. The cell suspension was seeded in 75-cm^2^ flasks and incubated at 37 °C in 5% CO_2_. After 48 h of culture, the cells were washed twice with phosphate-buffered saline (PBS) to remove nonadherent cells. At 70–80% confluence, the cells were passaged with 0.05% trypsin. After 3 to 4 passages, the cells were observed to be homogenous. The cell stocks of passage 4 were stored in liquid nitrogen (1.5 × 10^6^ cells/mL/vial). The quality of the AMSCs was evaluated by performing tests for sterility, mycoplasma, cell viability, endotoxin, and viruses. The surface markers of the AMSCs were determined through quantitative polymerase chain reaction (qPCR) analysis of the gene expression of chicks CD34, CD44, CD71, and CD73.

### 4.2. Generation of Bio-Pulsed AMSCs Conditioned Media

To generate the control AMSC conditioned medium (CM, without bio-pulsed treatment) and bio-pulsed AMSC CM, a cell stock was thawed and sub-cultured until passage 7. The AMSCs at passage 7 were plated at a density of 5000 cells/cm^2^ and cultured in DMEM containing 5% FBS in a humidified atmosphere of 5% CO_2_ in air at 37 °C for 3 days up to 85% confluence. The cells (1 × 10^8^ cells in total) were washed three times with PBS and cultured in serum-free DMEM. Then, the cells were treated with or without the bio-pulsed reagent (10 μg/mL extract of *P. multiflorum* Thunb, which was kindly gifted by Professors Ching-Chiung Wang and Hung-Yun Lin of Taipei Medical University, Taipei City, Taiwan [61,62,63]) and cultured for 72 h. The control AMSC CM and bio-pulsed AMSC CM were collected from the cultured cells treated without and with the bio-pulsed reagent, respectively.

### 4.3. Isolation and Lyophilization of AMSC-sEVs

Bio-pulsed AMSC-sEVs and control AMSC-sEVs were isolated from the bio-pulsed AMSC CM and control AMSC CM, respectively, by using the tangential flow filtration (TFF) method with size exclusion chromatography. Briefly, to remove nonexosomal particles, including cells, cell debris, microvesicles, and apoptotic bodies, the control AMSC CM and bio-pulsed AMSC CM were first centrifuged at 1000× *g* for 10 min and then at 10,000× *g* for 10 min. Subsequently, the centrifuged control AMSC CM and bio-pulsed AMSC CM were filtered through a 0.22-μm polyethersulfone membrane (Merck Millipore, Billerica, MA, USA). The control AMSC CM and bio-pulsed AMSC CM were first concentrated through TFF using a hollow fiber membrane cartridge (MidiKros Hollow Filter Modules, Repligen Corporation, Waltham, MA, USA) with a molecular weight cutoff of 500 kDa, and buffer exchange was performed through diafiltration against PBS. Then, the qEV column (qEV10–35 nm) (Izon Science, Christchurch, New Zealand) was used to isolate control AMSC-sEVs and bio-pulsed AMSC-sEVs according to the manufacturer’s instructions. The isolated control AMSC-sEVs and bio-pulsed AMSC-sEVs were aliquoted into sterilized polypropylene disposable tubes and stored at −80 °C until lyophilization. The frozen control AMSC-sEVs and bio-pulsed AMSC-sEVs were then freeze dried at −51 °C using a FreeZone 2.5-L Benchtop Freeze Dryer (Labconco Corporation, Kansas City, MO, USA). The freeze-dried powders of the lyophilized control AMSC-sEVs and bio-pulsed AMSC-sEVs were weighted and reconstituted in sterilize ultrapure water (QH_2_O) for subsequent experiments. The characterization and profile analysis of AMSC-sEVs were conducted following the Minimal Information for Studies of Extracellular Vesicles 2018 (MISEV2018) recommended by the International Society of Extracellular Vesicles [64].

### 4.4. Transmission Electron Microscopy

The frozen control and bio-pulsed AMSC-sEVs (unlyophilized) were maintained at 4 °C until completely thawed. The lyophilized bio-pulsed AMSC-sEVs were reconstituted in sterilize QH_2_O. All the aforementioned AMSC-sEVs were prepared for transmission electron microscopy (TEM), as described previously [65]. Briefly, the AMSC-sEVs samples were mounted on copper grids, fixed using 1% glutaraldehyde in cold PBS for 5 min to stabilize the immunoreaction, washed in sterile distilled water, contrasted with uranyl oxalate solution at pH 7 for 5 min, and embedded using methyl cellulose-UA for 10 min on ice. Excess cellulose was removed, and the samples were dried for permanent preservation. A Hitachi HT7700 transmission electron microscope (Tokyo, Japan) was used to image sEV samples at a voltage of 80 kV.

### 4.5. Particle Size Distribution and Concentration Measurements

To measure the particle size and concentration of the control and bio-pulsed AMSC-sEVs, tunable resistive pulse sensing (TRPS) measurements were performed using Exoid’s TRPS measurement system (IZON Science, New Zealand). Eighteen milligrams of the lyophilized control and bio-pulsed AMSC-sEVs were reconstituted in 500 μL of PBS, and the detailed procedure has been reported previously [66,67].

### 4.6. Western Blot Analysis

Exosomal proteins were separated through sodium dodecyl sulfate (SDS)–polyacrylamide gel electrophoresis and transferred onto polyvinylidene fluoride membranes (Millipore Corp., Bedford, MA, USA). The membranes were blocked in Tris-buffered saline (TBS) containing 5% nonfat milk at room temperature for 2 h and then incubated with specific primary antibodies at 4 °C overnight (CD9, CD81, TSG101, and Calnexin; 1:1000, GeneTex International Corporation, Hsinchu City, Taiwan) (CD63; 1:1000, iReal Biotechnology, Inc., Hsinchu City, Taiwan). The membranes were washed six times with TBST (TBS containing 0.1% Tween-20) and then incubated with a peroxidase-labeled anti-rabbit or anti-mouse secondary antibody (1:4000, GeneTex International Corporation). After washing six times with TBST, the membranes were visualized through chemiluminescence using the enhanced chemiluminescence (ECL) Western blot analysis system (Novex ECL Chemiluminescent Substrate, Thermo Fisher Scientific Inc.). Images of the blots were visualized and recorded using ChemiDoc XRS^+^ Imaging Systems with Image Lab Software (Bio-Rad Laboratories, Inc., Hercules, CA, USA).

### 4.7. Protein Contamination Assay

A Pierce BCA protein assay kit (#23225; Thermo Fisher Scientific) was used for the purity test. A standard curve (range: 0–2000 μg/mL) was plotted using nine points with serial dilution of the samples with bovine serum albumin (BSA) and a working reagent. All the samples and standard points were replicated three times. The samples (10 μL of each from the control AMSC CM and bio-pulsed AMSC CM and 10 μL of each from the lyophilized control AMSC-sEVs (100 mg) and bio-pulsed AMSC-sEVs (100 mg) were individually dissolved in 500 μL of QH_2_O) were mixed with 200 μL of the working reagent and incubated at 65 °C for 30 min. After cooling to room temperature, the absorbance difference of each sample, which came from each absorbance of sample subtracted by the averaged absorbance of blank standard replicates at 562 nm, was measured using a VersaMax ELISA microplate reader (Molecular Devices, San Jose, CA, USA), and the absorbance differences were converted into microgram per milliliter using the standard curve. If a protein concentration exceeded the upper limit of 2000 μg/mL of the standard curve, the sample was diluted until it could be measured within the standard range, and the final concentrate was calibrated considering the dilution factor.

### 4.8. RNAse Protection Assay

The lyophilized bio-pulsed AMSC-sEVs (contained 1.0 × 10^10^ particles on average) were weighted and resuspended and then incubated with or without RNase A (final concentration 100 ng/mL; Thermo Fischer Scientific, Waltham, MA, USA) at 37 °C for 10 min, as described earlier [68]. Finally, RNA was extracted and profiled by using the Agilent 2100 Bioanalyser for small RNA profiles with the Small RNA kit (Agilent Technologies, Santa Clara, CA, USA).

### 4.9. Exosomal Protein Concentration Examination

The lyophilized control AMSC-sEVs and lyophilized bio-pulsed AMSC-sEVs (both contained 1.0 × 10^11^ particles on average) were individually weighted and resuspended in RIPA buffer (50 mM Tris-HCl (pH 7.4), 150 mM NaCl, 1% Triton X-100, 1% Na-deoxycholate, 0.1% SDS, 0.1 mM CaCl_2_, and 0.01 mM MgCl_2_) supplemented with protease inhibitor cocktail (Thermo Fisher Scientific). The Pierce BCA protein assay kit (#23225; Thermo Fisher Scientific) was used for the purity test. A standard curve (range: 0–250 μg/mL) was plotted using seven points of serial dilution with BSA and a working reagent. All the samples and standard points were replicated three times. The samples (10 μL each) were mixed with 200 μL of the working reagent and incubated at 65 °C for 30 min. After cooling to room temperature, the absorbance difference of each sample, which came from each absorbance of sample subtracted by the averaged absorbance of blank standard replicates at 562 nm, was measured using a VersaMax ELISA microplate reader (Molecular Devices), and the absorbance difference was converted into microgram per milliliter using the standard curve.

### 4.10. Exosomal RNA Concentration Examination

The lyophilized control AMSC-sEVs and lyophilized bio-pulsed AMSC-sEVs (both contained 1.0 × 10^11^ particles on average) were individually weighted and resuspended in lysis buffer A of the qEV RNA extraction kit (IZON Science). Extraction was performed according to the manufacturer’s instructions. The extracted exosomal RNA concentration of the control AMSC-sEVs and bio-pulsed AMSC-sEVs was determined using a NanoDrop 2000c Spectrophotometer (Thermo Fisher Scientific).

### 4.11. Cell Proliferation Assay (MTS Assay)

To determine cell viability, human skin fibroblasts (HSFs) and human follicle dermal papilla cells (HFDPCs) were purchased from the Bioresource Collection and Research Center of the Food Industry Research and Development Institute (Hsinchu City, Taiwan) and PromoCell GmbH (Heidelberg, Germany), respectively. The experimental protocol was approved by the Institutional Review Board of the Tri-Service General Hospital and National Defense Medical Center (TSGHIRB No. E202216027). The AMSCs, HSFs, and HFDPCs were cultured in their growth media until they reached 80% confluence. The cells were seeded in 96-well plates at a density of 5 × 10^3^ cells/100 mL per well. After seeding, the cells were incubated at 37 °C in 5% CO_2_ for 24 h to allow for cell attachment. The cells were then washed once with PBS, and the medium was replaced with the medium containing 0.25% stripped FBS for starvation overnight. The medium was replaced, and the AMSCs were treated with the PM extract (0, 0.04, 0.4, 4, 10, 20, 30, and 40 μg/mL, dissolved in DMSO) for 4 days; the vehicle control (0 μg/mL) was added at an equal volume to the DMSO. For the HSFs and HFDPCs, the medium was supplemented with the reconstituted control AMSC-sEVs (70 μg/mL) and bio-pulsed AMSC-sEVs at different concentrations (0, 0.07, 0.7, 3.5, 7, 35, 70, and 350 μg/mL, dissolved in PBS) for treatment for 4 days; the vehicle control (0 μg/mL) was added at an equal volume to the PBS. The PM extract- and AMSC-sEV-medium mixtures were refreshed every other day. After 4 days of treatment, the growth medium was replaced with the medium containing 20% MTS solution (CellTiter 96 AQueous One Solution Cell Proliferation Assay Kit; Promega, Madison, WI, USA), with incubation for 2 h. The absorbance of each well at 490 nm was measured using a VersaMax ELISA microplate reader (Molecular Devices).

### 4.12. Wound Healing Assay

To examine the wound healing capability of the HSFs after treatment with the control AMSC-sEVs (70 μg/mL, dissolved in PBS) and bio-pulsed AMSC-sEVs (7, 35, 70, and 140 μg/mL, dissolved in PBS), wound healing experiments were performed using a silicone insert (culture insert, 2-well plate, ibiTreat; Ibidi, Martinsried, Germany), with the cells being seeded in a 24-well plate. Specifically, the cells were seeded at a density of 2.5 × 10^5^ cells/cm^2^ in two compartments of the silicone insert. The cells were grown for 24 h, after which the medium was replaced with starvation medium (serum free), and the cells were cultured for another 24 h. The culture inserts were removed using sterile tweezers, resulting in a 500-μm-wide gap. The medium supplemented with 0.25% stripped FBS and the control AMSC-sEVs, bio-pulsed AMSC-sEVs at different concentrations, or the vehicle control (0 μg/mL, the equal volume of PBS) was added, with further incubation for 24 h

The subsequent healing process in one well was recorded using an Olympus IX50 inverted microscope (Hamburg, Germany). In addition, the images of the starting conditions (500-μm gaps) were captured for at least two to three wells; all the wells of the 24-well plate were evaluated visually for any anomalies. After the wounding process was performed for 24 h, the micrographs of all the wells were captured using an Olympus IX50 inverted microscope.

### 4.13. Quantitative Real-Time Polymerase Chain Reaction

To examine gene expression in the AMSCs, HSFs, and HFDPCs, we conducted qRT-PCR. The AMSCs were starved and treated with the PM extract (0, 0.04, 0.4, 4, 10, and 20 μg/mL) for 72 h. The HSFs and HFDPCs were starved and then treated with medium supplemented with 0.25% stripped FBS and the control AMSC-sEVs, and bio-pulsed AMSC-sEVs at different concentrations for 2 days. To examine the anti-inflammatory effect, the HSFs were starved and then treated with 1 μg/mL of *Escherichia coli* 0111:B4 lipopolysaccharide (LPS) (L4391, MilliporeSigma, Burlington, MA, USA), control AMSC-sEVs (70 μg/mL), and bio-pulsed AMSC-sEVs at different concentrations (0.07, 0.7, 3.5, 7, 35, 70, and 350 μg/mL) for 6 h. The cells were then collected and subjected to qRT-PCR. Total RNA was extracted using a GENEzol TriRNA Pure Kit (Geneaid Biotech Ltd., New Taipei City, Taiwan) after the removal of genomic DNA; cDNA was synthesized using 1 μg of DNase I-treated total RNA by employing the RevertAid H Minus First Strand cDNA Synthesis Kit (Thermo Fisher Scientific). The prepared cDNA was used as the template for qRT-PCR, which was conducted using the QuantiFast SYBR Green PCR Kit (Qiagen, Hilden, Germany) on a Rotor-Gene Q Real-Time PCR Detection System (Qiagen). The qRT-PCR process involved initial denaturation at 95 °C for 5 min, followed by 45 cycles of denaturation at 95 °C for 10 s and combined annealing and extension at 60 °C for 30 s according to the manufacturer’s instructions. *Gallus gallus* and *Homo sapiens* primer sequences are listed in Table 1. Calculations of the relative gene expression (normalized to *gapdh* and *18S* as reference genes) were performed using the 2^−ΔΔCT^ method. qRT-PCR fidelity was determined by examining the melting temperature.

### 4.14. Statistical Analysis

Data on cell viability, gene expression, sEV particle number, wound healing cell number, protein, and exosomal protein/RNA concentration was analyzed using IBM SPSS Statistics version 19.0 (SPSS Inc., Chicago, IL, USA). One-way analyses of variance (ANOVA) and two-tailed Student’s *t* test were conducted. The results were considered significant when the *p* value was <0.05 (*, # or $), <0.01 (**, ##, or $$), <0.001 (***, ###, or $$$).

## Figures and Tables

**Figure 1 ijms-23-15010-f001:**
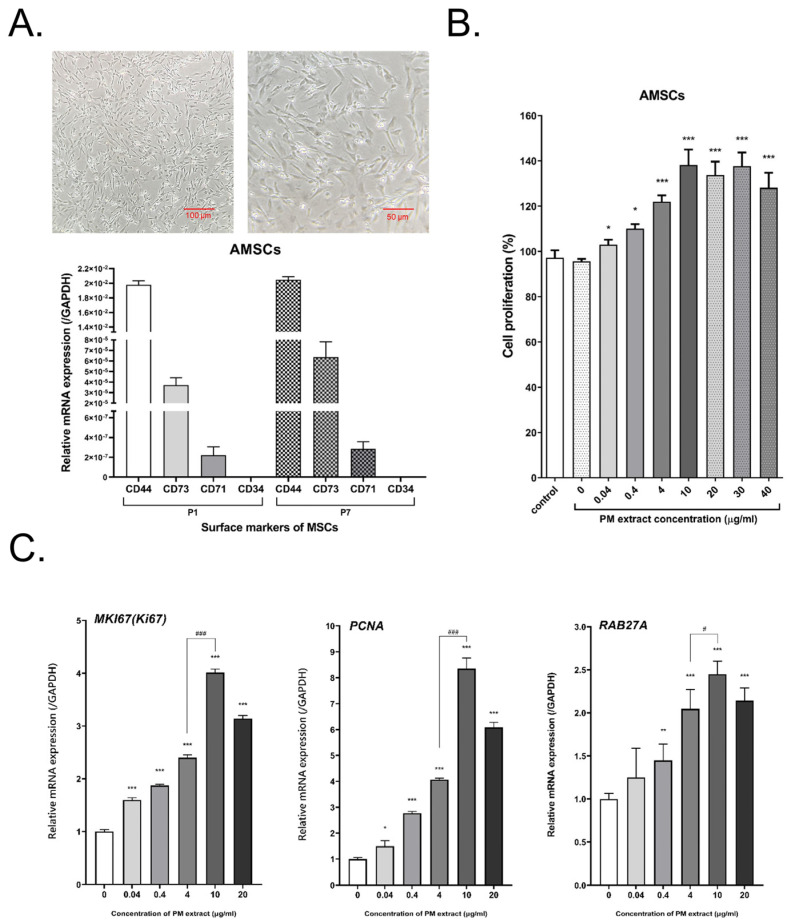
Avian−derived mesenchymal stem cell (ASMC) characteristics and effect of the *Polygonum multiflorum* (PM) extract on AMSCs. (**A**) The morphology and surface markers of AMSCs. (Bar = 100 μm). (**B**) Treatment with the PM extract enhanced cell proliferation in AMSCs. (Number of repeats n = 4, data are expressed as the mean ± standard deviation; * *p* < 0.05, *** *p* < 0.001, compared with the control group). (**C**) Treatment with the PM extract enhanced *MKI67*, *PCNA*, and *RAB27A* expression in AMSCs. (n = 4, data are expressed as the mean ± standard deviation; * *p* < 0.05, ** *p* < 0.01, *** *p* < 0.001, compared with 0 μg/mL; # *p* < 0.05, ### *p* < 0.001, compared with 4 μg/mL).

**Figure 2 ijms-23-15010-f002:**
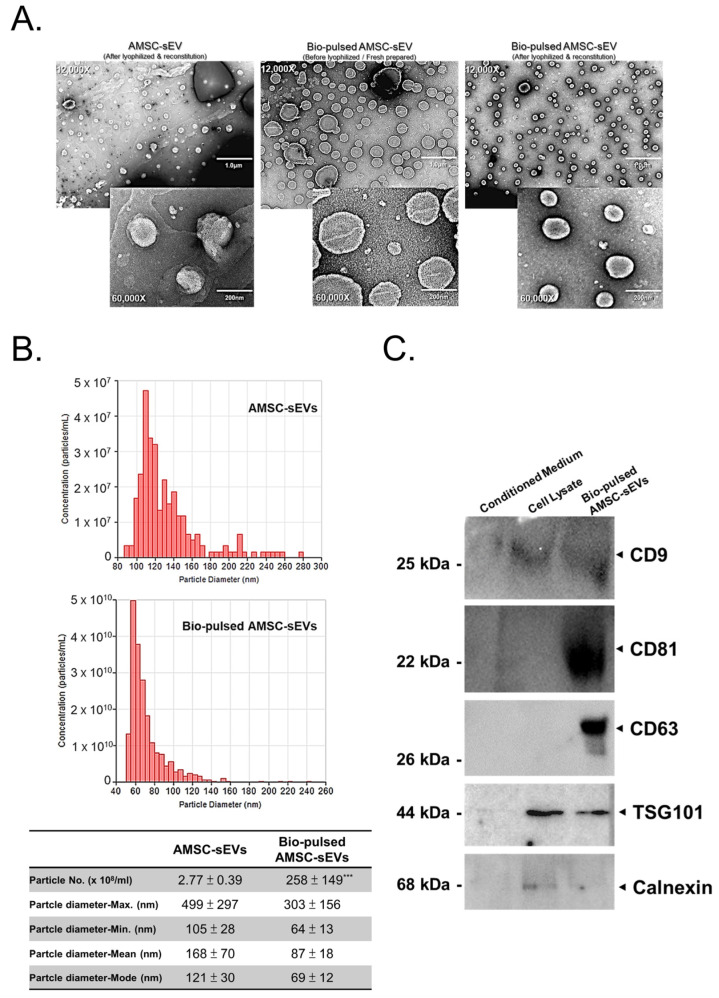
Characteristics and advantages of bio-pulsed AMSC-sEVs. (**A**) Images of sEVs were obtained through transmission electron microscopy. Morphology did not alter after lyophilizing and reconstituting. (**B**) The particle size distribution and number were analyzed by the tunable resistive pulse sensing measurement system. (n = 3, data are expressed as the mean ± standard deviation; *** *p* < 0.001, compared with the AMSC-sEVs). (**C**) The extracellular vesicle markers of CD9, CD63, CD81, TSG101, and a contamination marker, calnexin, were examined through Western blot.

**Figure 3 ijms-23-15010-f003:**
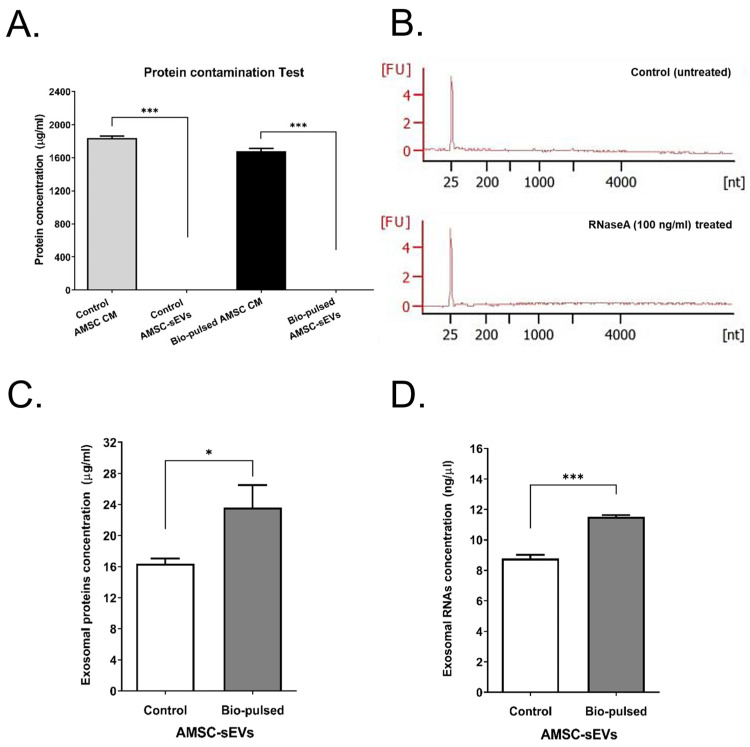
The quality of bio-pulsed AMSC-sEVs. (**A**) Residual proteins contained in lyophilized powder (outside the exosomes) were almost removed through purification and isolation. (n = 3, data are expressed as the mean ± standard deviation; *** *p* < 0.001, compared with the control AMSC CM or bio-pulsed AMSC CM). (**B**) Small RNA profiles extracted from bio-pulsed AMSC-sEVs and after RNaseA (100 ng/mL) treatment. RNA was extracted from samples and run on a Small RNA Bioanalyzer assay. The concentrations of exosomal proteins (**C**) and RNAs (**D**) were analyzed after lysing and extracting from both the lyophilized sEVs. (n = 3, data are expressed as the mean ± standard deviation; * *p* < 0.05, *** *p* < 0.001, compared with the control AMSC-sEVs).

**Figure 4 ijms-23-15010-f004:**
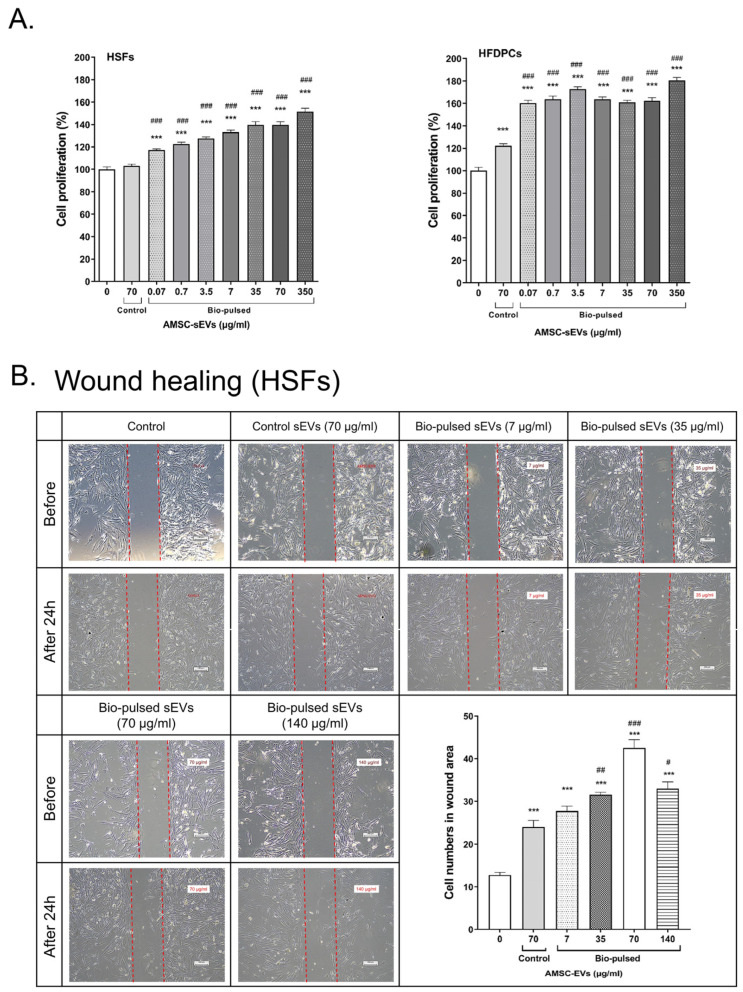
Bio-pulsed avian-derived mesenchymal stem cell (AMSC)-sEVs enhanced cell proliferation and wound healing. (**A**) The cell proliferation of human skin fibroblasts (HSFs) and human follicle dermal papilla cells was significantly augmented after treatment with bio-pulsed AMSC-sEVs for 4 days. (n = 12, data are expressed as the mean ± standard deviation; *** *p* < 0.001, compared with the 0 μg/mL group; ### *p* < 0.001, compared with the control AMSC-sEV group). (**B**) Bio-pulsed AMSC-sEVs enhanced wound healing compared with control AMSC-sEVs in HSFs after treatment for 24 h. (n = 4, data are expressed as the mean ± standard deviation; *** *p* < 0.001, compared with the # *p* < 0.05, ## *p* < 0.01, ### *p* < 0.001, compared with the control AMSC-sEVs group).

**Figure 5 ijms-23-15010-f005:**
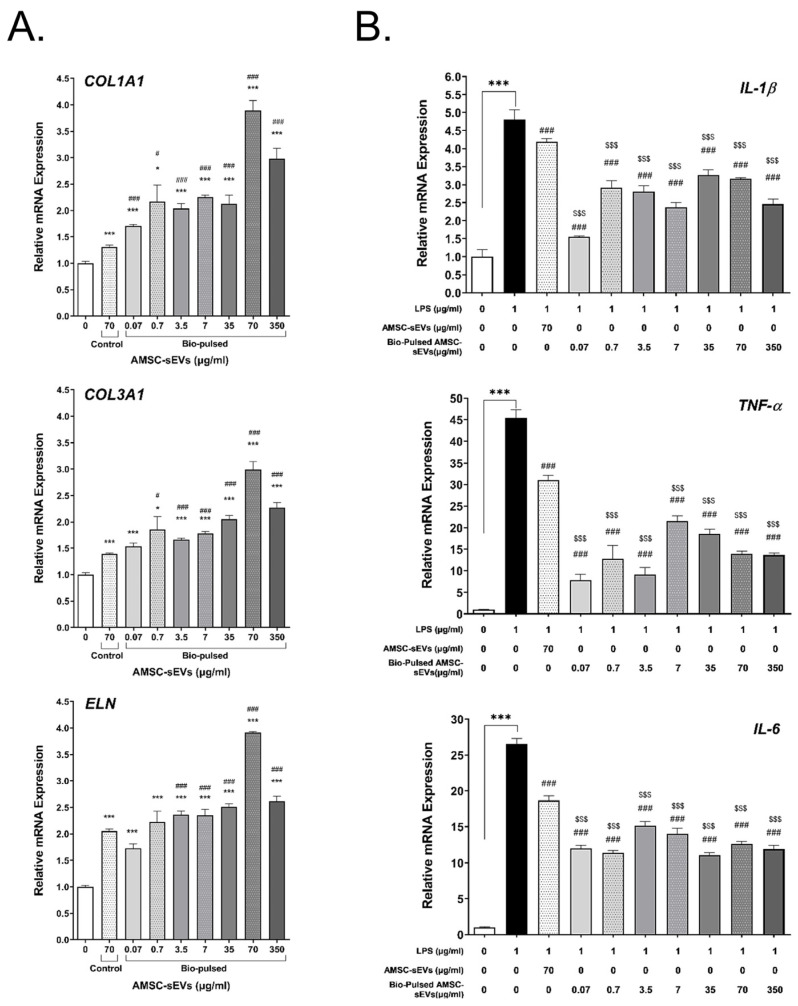
Bio-pulsed avian-derived mesenchymal stem cell (AMSC) sEVs enhanced collagen and elastin expression but ameliorated lipopolysaccharide (LPS)-induced inflammation on human skin fibroblasts (HSFs). (**A**) Bio-pulsed AMSC-sEVs upregulated the expression of *COL1A1*, *COL3A1*, and *ELN* compared with control AMSC-sEVs in HSFs after treatment for 2 days. (n = 4, data are expressed as the mean ± standard deviation; * *p* < 0.05, *** *p* < 0.001, compared with 0 μg/mL; # *p* < 0.05, ### *p* < 0.001, compared with the control AMSC-sEVs group). (**B**) After treatment with control AMSC-sEVs or bio-pulsed AMSC-sEVs for 24 h, LPS-induced expression of cytokines, interleukin (IL)-1β, tumor necrosis factor-α, and IL-6 were ameliorated. (n = 4, data are expressed as the mean ± standard deviation; *** *p* < 0.001, compared with 0 μg/mL; ### *p* < 0.001, compared with the LPS group; $$$ *p* < 0.001, compared with the control AMSC-sEVs group).

**Figure 6 ijms-23-15010-f006:**
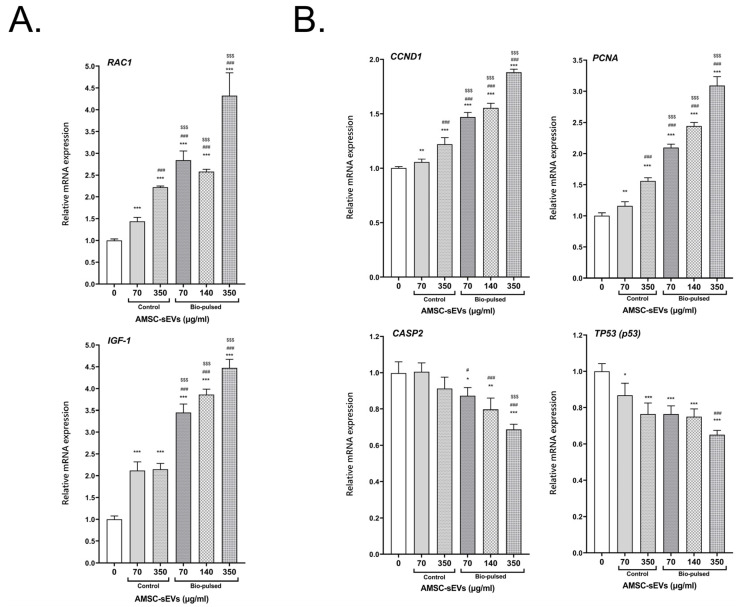
Bio-pulsed avian-derived mesenchymal stem cell (AMSC) sEVs improved hair follicle activity. (**A**) The hair follicle activity-associated genes *RAC1* and *IGF-1* were significantly augmented after treatment with bio-pulsed AMSC-sEVs for 2 days. (**B**) The cell proliferation of follicle dermal papilla cells was enhanced through the upregulation of the proliferative genes *CCND1* and *PCNA* and the inhibition of the apoptotic genes *CASP2* and *TP53* after treatment with bio-pulsed AMSC-sEVs for 2 days. (n = 4, data are expressed as the mean ± standard deviation; * *p* < 0.05, ** *p* < 0.01, *** *p* < 0.001, compared with 0 μg/mL; # *p* < 0.05, ### *p* < 0.001, compared with 70 μg/mL of control AMSC-sEVs group; $$$ *p* < 0.001, compared with 350 μg/mL of control AMSC-sEVs group).

**Table 1 ijms-23-15010-t001:** The primer sequences used for QPCR analysis.

**For *Gallus gallus***			
**Gene name**	**Forward**	**Backward**	**Accession No.**
** *CD44* **	CCGGGCTTTTCTTCCTTCTG	AGTTGGCCATTGTTTCCTCAG	NM_204860.5
** *CD73 (NT5E)* **	CTCCCGTTTCAAGGGTCAGG	TTCATGGTTGCCCAAAGCCA	XM_040669143.1
** *CD71 (TFRC)* **	TTGGAGACTCCTGATGCTATCG	TCACATAGACAGGTTTGCCAGA	NM_205256.2
** *CD34* **	TGGGAGTAAGAGTGGGGTCG	CCTGGAGCAGAAAAGGGACTT	XM_040691711.1
** *MKI67 (Ki-67)* **	AAACGGAATGGGACTGACGG	TCACATTCTGTTCTCCTTCCAAA	XM_040674390.2
** *PCNA* **	CGGATACGTTGGCTCTAGTGT	GGAATTCCAAGCTGCTCCAC	NM_204170.3
** *RAB27A* **	AGAAAAAGGCAAATGTGGCTGC	GAGAGTTCACTAAGGCTGCATGA	FJ449551.1
** *GAPDH* **	GTCAAGGCTGAGAACGGGAA	GCCCATTTGATGTTGCTGGG	NM_204305.2
**For *Homo sapiens***			
**Gene name**	**Forward**	**Backward**	**Accession No.**
** *COL1A1* **	GTCAGATGGGCCCCCG	CACCATCATTTCCACGAGCA	NM_000088.4
** *COL3A1* **	GAGGATGGTTGCACGAAACAC	CAGCCTTGCGTGTTCGATATT	NM_000090
** *ELN* **	CAGGTGCGGTGGTTCCTC	CTGGGTATACACCTGGCAGC	M36860
** *IL-1β* **	GCAGCCATGGCAGAAGTACC	AGTCATCCTCATTGCCACTGTAAT	NM_000576.2
** *TNF-α* **	TAGCCCATGTTGTAGCAAACCC	TTATCTCTCAGCTCCACGCCA	NM_000594.3
** *IL-6* **	ACCCCCAGGAGAAGATTCCA	GATGCCGTCGAGGATGTACC	M54894.1
** *RAC1* **	TGGCTAAGGAGATTGGTGCTG	CGGATCGCTTCGTCAAACAC	AF498964.1
** *IGF-1* **	ATCAGCAGTCTTCCAACCCAAT	GCCAGGTAGAAGAGATGCGA	M29644.1
** *CCND1* **	ATCAAGTGTGACCCGGACTG	CTTGGGGTCCATGTTCTGCT	NM_053056.3
** *PCNA* **	TCTGAGGGCTTCGACACCTA	TCATTGCCGGCGCATTTTAG	BC062439.1
** *CASP2* **	GCATGTACTCCCACCGTTGA	GACAGGCGGAGCTTCTTGTA	NM_032982.3
** *TP53* **	AAGTCTAGAGCCACCGTCCA	CAGTCTGGCTGCCAATCCA	NM_000546.5
** *18s* **	GTAACCCGTTGAACCCCATT	CCATCCAATCGGTAGTAGCG	NR_003286

Abbreviation: *NT5E*: 5′-nucleotidase ecto; *TFRC*: transferrin receptor; *PCNA*: proliferating cell nuclear antigen; *GAPDH*: glyceraldehyde-3-phosphate dehydrogenase. *COL1A1*: Human Collagen type I α1; *COL3A1*: Human Collagen type III α1; *ELN*: Human Elastin; *IL-1β*: interleukin 1 β; *TNF-α*: tumor necrosis factor α; *IL-6*: interleukin 6; *IGF-1*: insulin-like growth factor I; *CCND1*: cyclin D1; *PCNA*: proliferating cell nuclear antigen; *CASP2*: *TP53*: tumor protein p53. Symbols for genes are italicized.

## Data Availability

Data are contained within the articles.

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
