# Peer review of "Bio-Pulsed Stimulation Effectively Improves the Production of Avian Mesenchymal Stem Cell-Derived Extracellular Vesicles That Enhance the Bioactivity of Skin Fibroblasts and Hair Follicle Cells"

_ijms, 2022, doi:10.3390/ijms232315010_

Round 1
Reviewer 1 Report (Previous Reviewer 1)
N/A
Author Response
Thank you for the reviewer’s review.
Reviewer 2 Report (New Reviewer)
This study investigates the beneficial effect of avian MSC pre-conditioning with bio-pulsed reagents on extracellular vesicles bioactivity. Authors also studied the activity of bio-pulsed MSCs-derived EVs on skin fibroblast and hair follicle cells. Generally speaking, the study seems interesting, but it is not efficiently presented. The manuscript is written in a good English, but the organization of the article must be improved. There are some issues to address.
1. Authors used tangential flow filtration with size exclusion chromatography to isolate EVs from bio-pulsed MSCs conditioned medium. Although this method allows to obtain a cleaner EV product compared to the more commonly used differential ultra-centrifugation, there are still no methods allowing to perfectly isolate a pure population of exosomes. For this reason, it would be preferable to refer to them with a more general term “extracellular vesicles” rather than “exosomes”, in the manuscript.
2. Introduction
2.1. In the introduction the main concepts are reported in a not very linear and clear way. A main hypothesis is missing, and the main goal of the entire work is not well explained. This part should be improved by giving more connections between the different points.
2.2. Introduction does not highlight the novelty of the work, which seems to be represented by the bio-pulsing with PM. Why did authors choose this type of pre-conditioning and why this should be preferable compared to other type of cell priming? Which is the unfilled research gap? Which is the add-on value of present study? Please clarify and highlight it.
2.3. Authors used avian derived mesenchymal stromal cells, but the reason of this choice is clarified only in the discussion, where it is reported that “the use of cells, tissue, or products of human origin in cosmetic products is prohibited”. This evaluation could be stated directly in the introduction.
3. Statistics: as reported in methodologies, t-test was performed for all the analysis. However, t-test can be used only for comparisons between two groups. When the experimental set up consists of three or more experimental groups, a one-way ANOVA should be carried out. Please check statistics and modify when it is needed (in particular for data reported in Fig. 1B-C, Fig. 4, Fig. 5, Fig. 6).
4. Results
4.1. Figure 2A-B: data from TRPS in figure 2B reported that size of bio-pulsed AMSC-EVs was smaller compared to the size of control AMSC-EVs. However, TEM micrographs reported in figure 2A suggest an opposite result, in which bio-pulsed AMSC EVs are bigger than control. Are the images reported in the same magnification? Is it possible that images have been exchanged by mistake?
4.2. Figure 2C: western blot only reports data on bio-pulsed AMSC-EVs. Analysis on control untreated AMSC-EVs should be also performed.
4.3. Line 152: “in order to examinate the residual protein contained in the lyophilized exosomes.” maybe there is a typo, putting a dot instead of a comma.
4.4. In Figure 2 and Figure 3, authors reported that lyophilization does not affect the morphology of EVs and the miRNA integrity. However, it is not clear which would be the advantage of lyophilization: long-term storage? Stability? Please clarify.
4.5. Which type of EVs were used for experiments reported in Figure 4, 5 and 6? Lyophilized or not lyophilized and why? Please clarify.
4.6. Figure 5: these data seem interesting, but the rationale is not well explained. Please clarify and give more background information. Also, in this figure some texts are hard to read.
5. Discussion is just a summary of the results, with an initial background section. Authors should provide a better discussion of the data, considering also what is reported in the literature.
Round 2
Reviewer 2 Report (New Reviewer)
Changes performed by authors improved the quality of the manuscript, which can be accepted in the present form.
This manuscript is a resubmission of an earlier submission. The following is a list of the peer review reports and author responses from that submission.
Round 1
Reviewer 1 Report
The authors of the ijms-1880970 manuscript claim to have used a bioreagent to stimulate the production of exosomes by avian mesenchymal stem cells, and then use these exosomes to improve the bioactivity of cells related to skin and hair. I have important conceptual and methodological concerns about the study that should be addressed before it could be considered for publication.
MAJOR COMMENT
According to the MISEV2018 guidelines, at least one protein marker in each of three categories should be presented to confirm extracellular vesicle nature and purity (transmembrane proteins associated with the plasma membrane and/or endosomes, cytosolic proteins, and non-EV co-isolated structures). The authors only used CD9 and CD81 (transmembrane), which can only prove that their sample contained lipid bilayers (not necessarily intact or enclosed vesicles). Therefore, using the term “exosome” or even “extracellular vesicles” is not appropriate in this case. The authors used conventional TEM, which does not show the intact lipid bilayer structure of the EVs, so the samples could just contain another kind of extracellular particles instead of extracellular vesicles. The authors should provide further protein characterization of their samples or change the focus of the study to something like “non-vesicular extracellular particles”.
INTRODUCTION
· Lines 56-57: please revise the term “endemic species” since its meaning is “plants and animals that exist only in one geographic region”, which does not seem to be what the authors wished to say.
· Please elaborate more in the introduction on how you reached the hypothesis that exosomes could be the bioactive agents of your system and not the AMSCs themselves or their conditioned medium, for example. Please add the rationale and literature evidence that pointed in this direction.
· In the same way, please provide literature support for the idea that P. multiflorum Thunb extract would stimulate the release of extracellular vesicles.
· Please define in the introduction what you mean by bio-pulsed as it is not a common term in the scientific literature.
RESULTS
· In section 2.3, please use either “proliferation” or “viability” in the text and figure since mixing both terms is difficult reading.
· In Figure 4B, there is no evident change in wound healing for any of the experimental groups. There are a bit more cells in the open area of bio-pulsed exo after 24 hours, but this likely results from the fact that a few cells remained in that are in the beginning as can be seen in the “before” image.
METHODS
· For all experiments, please provide information about how many times they were repeated independently and the number of technical replicates.
· Please provide references from the literature showing that gene expression of CD34, CD44, CD71, and CD73 is enough for validating the cells as AMSCs. If no support is available, could the authors please provide protein quantification data (western blotting, ELISA, flow cytometry)?
· Related to that, could you please provide data showing that the AMSCs are still AMSCs 7 passages after their characterization and freezing?
· Please provide information about the P. multiflorum Thunb extract. Source material, how was it prepared, quality controls, what kind of extract it is, chemical characterization, etc., so that the study can be reproduced by others. Also, please describe what was considered the vehicle control group.
· Do you have data showing the viability of the AMSCs after being cultured for 72 hours in serum-free medium? This is a long time, and could significantly impact cells viability, which can significantly impact extracellular vesicles release and purity.
· Please provide more information about the qEV column used, especially size range.
· In the experiments using the HSFs and HFDPCs, were they cultured in the presence of FBS? Extracellular vesicles present in FBS could mask the effects of extracellular vesicles in the tested samples.
· Are HSFs known for responding to LPS? Please provide some explanation of why you used this condition in the gene expression experiments and references that support it.
DISCUSSION
· Line 242: I suggest changing the statement for something like “The bio-pulsed AMSC exosomes (please reconsider this term) exhibited bioactivity with potential implications for skin and hair aesthetics.” as you used in vitro cellular models that are limited in representing effects on skin or hair tissues.
Reviewer 2 Report
Within the article "Bio-pulsed stimulation effectively improves the production of avian mesenchymal stem cells-derived extracellular vesicles that enhance the bioactivity of skin and hair" the authors decribe their findings of the effect of activating avian mesenchymal stem cells (AMSC, isolated from chicken embryos) with Polygonum multiflorum (PM) extracts on the exosome secretion. Further, the authors showed an enhanced bioactivity (based on proliferation, marker gene expression and immune modulation) of human skin fibroblasts (HSFs) and human folicle dermal papilla cells (HFDPCs) after exposition to bio-pulsed AMSC-derived exosomes. This may serve as a starting point in utilization of AVMSC-derived exosomes in medical application or aesthetics. Although the overall study is well described and the concept of utilzing PM bio-pulsed exosomes in priming/modulating targeted cells can be easily followed by the reader, some concerns need to be adressed in a major revision.
Major concerns:
1. The authors claim, that AMSC-derived exosomes enhance the bioactivity of skin and hair (already in the title!). However, only the effects on HSFs and HFDPCs were tested within this study. Further, the effects were only tested for selected read outs (proliferation, marker genes etc.). Thus, as skin and hair consists of more than HSFs and HFDPCs, the claim of an overall benefit on skin and hair need to be adressed by the author in the discussion section. Further, the reviewer highly recomments to change the title of the manuscript accordingly.
2. The rationale behind the usage of embryonic chicken-derived MSC for the study is not clear. In the introduction, the authors describe the progress in utilizing dermal-derived human, murine or rabbit MSCs as ideal source for for tissue engineering (ll. 50-55). Although displaying this ideal source, the authors name chicken as suitable animal model due to their abundant dermis (without providing any reference) and listing global economy as reason (ll. 55-57). It is not clear if the statement of the abundant dermis is refered to adult chicken. And still, without any explanation, embryonic chicken MSCs were used in the study.
3. Refering to point 2: What is the overall aim of the study? What is the impact of the study on the field? It is not clear if embryonic chicken MSC only serve as a model for this basic study or if they should be directly utilized in medical or aesthetics applications. The aim need to be adressed in the introduction, whereas the impact need to be discussed in the discussion section.
4. Refering to point 3: If embryonic chicken MSCs should be directly used in human applications, are there any host responses or transmissions of infections to be considered? Please include this into the discussion section.
5. The authors used a lyphilization as formulation for their exosomes. However, it is not clear if and how this affects the activity of exosomes. The authors only provided transmision electron microscopy results. Comparative analysis especially of the biocativity of freshly isolated, frozen and lyphilized exosomes should be included.
6. The context of the study is insufficiently described. It is not known, if any of the findings were previously reported for human or rodent studies. Are their any mechanisms known, by which MSC-derived exosomes enhance the acivity of fibroblats? Are there any mechanisms known how PM-bio-pulsing affects MSCs?
Minor concerns:
1. The progress of dermis-derived separation, culture and differentiation stated in the introduction (ll. 50-53) is insufficiantly explained and not provided with any references. It would be great to state the most common techniques.
2. The authors state, that MSCs among other cell types, produce the highest amount of EVs (63-63). There is no refecence giving. Is this also true in comparison to activated immune cells?
3. Most bar charts are provided as coloured. Is this really necessary? Please consider grey scales or black/white graps.
5. In section 2.2 (ll. 112) the authors claim, that the quality of AMSC-derived exosomes was increased by bio-pulsing. However, it is not clear what the read-out for "qualtiy" is.
6. In ll. 116-117, the authors state, that the particle size distribution and number of the exosomes differed after bio-pulsed treatment, but not significantly. What is meant by this? With regard to the table in Figure 2B, a huge difference of 3 orders of magnitude in number of particles is observed prior and after bio-pulsing.
7. Why were the surface markers CD9 and CD81 only analyzed for bio-pulsed exosomes and not control exosomes (Figure 2C)?
8. In figure 3A, the authors determine the protein contamination of isolated exosomes. It is not clear what is meant by "protein contamination". Further, it is not clear, which fraction/purification step is used as "Unpurified AMSC-exosomes" and which fraction/purification step as "Purified AMSC-exosomes".
9. It is not clear how the exosome input was normalized for the experiments. The the method section (e.g. l. 374) the authors state, the the particle number per sample was known but were individually weighted. Why and how were the exosomes weighted although the particle number eas known? How were the inputs normalized for figure 3A and 3B? This is especially of importance, since in Figure 3B the authors state a higher protein and RNA contens for bio-pulsed exosomes.
10. In Figure 4B, the authors state an increased wound healing by HSF. Was there any quantitative read-out (e.g. cell numbers in the previously empty region) to proof this statement?
11. In figures 4-6, the authors observed significantly increased outcomes already by non-bio-pulsed exosomes and state, that this effects are further enhanced by bio-pulsed exosomes. Please provide statistical significance for this statements.
12. In ll. 167-168 the authors state to examine the anti-inflammatory activity of HSFs. However, only pro-inflammatory cytokines (IL1b, IL6 and TNFa) were analyzed. Do the authors mean the anti-inflammatory effects of bio-pulsed AMSC-derived exosomes on HSFs?
13. AMSCs were kept at FBS-free conditions for 72 h for exosomes collection from conditioned media. (section 4.2) Was the vaibility and bioactivity of cells checked compared to cells at normal culture conditions?
14. For different assays on the effects of AMSC-derived exosomes on HSFs and HFDPCs different dosings of exosomes were used (refer to 0, 0.07, 0.7, 3.5, 7, 35, 70, 350 µg/ml at line 409 and 7, 35, 70, 140 µg/ml at line 417). What is the reason?
Reviewer 3 Report
Submitted MS from Ju-Sheng Shieh proposed method to improve the quantity and biological properties of MSC however paper has lots of issues and cannot be accepted in current state. Following are my comments for the authors.
I. Figure 1.
· Confusing and mislabeled figure legends.
· I don’t understand how cell viability was correlated with cell proliferation. Wouldn’t simply cells count would be best way quantification. Higher MTS signal of assay could reflect higher mitochondrial activity
· Cellular morphology is hard to visualized.
· Why mRNA Rab27A was chosen as readout to predict favorable out of cells? What was the rationale and no other EV associated markers were considered?
II. Which component of Polygonum multiflorum (PM) extract is responsible for the phenotype? Protein or Chemicals or other alkaloids? Can any cell type exposed to Polygonum multiflorum (PM) extract will produce the bio-active EVs?
III. Use of terminology as Exosomes instead of EV is not correct as the preparation of EVs are pool of both Exo and ectosomes.
IV. Fig 2C
o Why non-pulsed AMSC derived exo were not tested for the markers?
o Expression of endosomal derived markers eg CD63 is missing.
o Include EV negeative markers calnexin and GM130
o Include loading control missing
V. Fig 3
· What it means to be called unpurified Vs purified Exo. This is super confusing and also not clear from the Fig legend or the text.
· What is starting volume of cell number from which EV samples were obtained?
· Was the RNA in EV samples protected? Consider conducting RNA protection assay
VI. Fig 4.
· Quantification of wound healing assay and statistic is missing?
· Is this assay wound healing or migration assay?
VII. Fig 5
· Why higher concentration (350) of bio-pulse Exo cause reduction in the TNF and IL-1B?
· Why the lower concentration (0.07) of bio-pulse EV cause reduction in TNF and IL-1B in LPS treated condition?
· No dose response trend were observed.
Round 2
Reviewer 1 Report
I appreciate the authors' responses and changes in the manuscript, which has been much improved. I only have a few last minor suggestions:
1. Please revise section 4.13 Statistical Analysis (where the authors say that a two-tailed Student’s t test was conducted) and the figure legends to make sure that you mention the correct analysis applied for each specific assay.
2. Please clarify in the manuscript that calnexin is a "contamination" marker, usually only present in large EV samples. Otherwise, it gives the impression that you used it as an exosome marker.
3. Please describe in the methods section what was used as vehicle control for all experiments.
Reviewer 2 Report
Thank you for thoroughly modifying your manuscript based on the first report.
Author Response
Comments and Suggestions for Authors
Thank you for thoroughly modifying your manuscript based on the first report.
Answer:
Thank you for the reviewer’s comments.
Reviewer 3 Report
Please find attached PDF file.
